# PC-JeDi: Diffusion for Particle Cloud Generation in High Energy Physics

Matthew Leigh, Debajyoti Sengupta, Guillaume Quétant, John Andrew Raine, Knut Zoch, and Tobias Golling,

University of Geneva

22$^{nd}$ February, 2024

## Abstract

In this paper, we present a new method to efficiently generate jets in High Energy Physics called PC-JeDi. This method utilises score-based diffusion models in conjunction with transformers which are well suited to the task of generating jets as particle clouds due to their permutation equivariance. PC-JeDi achieves competitive performance with current state-of-the-art methods across several metrics that evaluate the quality of the generated jets. Although slower than other models, due to the large number of forward passes required by diffusion models, it is still substantially faster than traditional detailed simulation. Furthermore, PC-JeDi uses conditional generation to produce jets with a desired mass and transverse momentum for two different particles, top quarks and gluons.

# 1   Introduction

In high energy physics (HEP) experiments operating at the energy and intensity frontier, such as the ATLAS and CMS experiments [1, 2], simulated proton-proton collision events play a crucial role in precision measurements and searches for new physics phenomena. One of the current challenges posed by the increasing data collected by these experiments is the required computing resources for detailed simulated collisions. As a result, attention has turned to the use of fast surrogate models to reduce the computational cost of both event and detector simulation.

An object of particular interest at hadron colliders are jets. Jets are reconstructed from the collimated shower of particles resulting from the hadronisation of a quark or gluon produced in collisions, and are one of the most computationally intensive objects to simulate. These jets are captured by particle detectors and are studied in detail as they carry information about the particle which initiated the jet and the underlying physics of the hadronisation process. In recent years, state-of-the-art algorithms for studying jets rely on highly accurate modelling of their internal structure and extensively use machine learning (see Refs. [3, 4]). As such, any fast surrogate model needs to be able to accurately reproduce the complex and stochastic substructure observed in jets.

In this work we introduce a novel method for generating jets as particle clouds using transformers trained to reverse a diffusion process, which we call PC-JeDi (Particle Cloud Jets with Diffusion). With PC-JeDi we can generate new jets by first sampling noise for the momenta of a set number of constituent particles, and applying subsequent denoising operations. We train two PC-JeDi models to generate jets with large transverse momentum arising from two vastly different elementary particles, top quarks and gluons. We evaluate the performance using a variety of metrics used by previous approaches, and in addition look at the ability to capture the distributions of commonly used substructure observables.

# 2   Related work

Fast surrogate and supplementary models have been an important area of study for particle physics experiments operating at the energy and intensity frontier [5–7]. Detailed simulation of both particle interactions as well as the detector response to incoming particles represents a significant computational cost and many approaches are considered to supplement or replace traditional methods.

Parametrised models have been studied as replacements for expensive Monte Carlo (MC) simulation and detailed detector simulation, but in recent years deep generative models using Generative Adversarial Networks (GANs), Variational Autoencoders (VAEs), and Normalising

Flows have been used for detector simulation [8–18], event simulation [19–30], and the generation of jet constituents [31,32]. Typically the particles and particle showers generated in these approaches are represented by images or ordered vectors, and as such are not preserve permutation invariance. Recently, instead of representing data as a structured grid or vector, point clouds have been used to represent the data for generating jets in particle detectors [33–36], which present a more natural way of describing the underlying processes.

Outside of HEP, diffusion and score-based models have recently been shown to achieve state-of-the-art performance [37–40], and these have now been applied to detector simulation [41] in high energy physics on image based representations. However, as yet they have not been applied to point cloud generation in HEP, despite success in other fields such as protein and molecular generation [42,43].

In this work, both point cloud representation of jets and diffusion models are combined to generate jets. Comparisons of performance are made between PC-JeDi and the MPGAN approach introduced in Ref. [33]. Both methods are trained on the same dataset, however in comparison to MPGAN, message passing graph layers are replaced with transformers and the GAN is replaced by a diffusion model.

## 3   Generative diffusion models

Generative Diffusion Models are a broad family of probabilistic models which learn to reverse a process in which data is progressively perturbed by the injection of noise. In the past couple of years these models have been very successful in image generation, overtaking GANs in generation fidelity [37,44]. One of the major strengths of diffusion models are the stability of the training process, especially in comparison to GANs. The diffusion process can be described in the context of score matching [45,46], where the training objective is to model the so-called score function of the data [47]. In the limit of an infinitesimal time step, the perturbation and denoising operations can be framed as solutions to a stochastic differential equation (SDE) [38].

We construct the forward diffusion process $\{x_t\}_{t=0}^1$ of a variable $x \in \mathbb{R}^d$, indexed by a continuous time variable $t \in [0,1]$. The boundary conditions are chosen such that at the start of the process points are drawn from the independently and identically distributed data distribution $x(t=0) \sim p_{\text{data}}$, while the final points follow some prior distribution $x(t=1) \sim p_{\text{prior}}$, which is chosen to be a multivariate standard normal distribution. We also denote $p(x_t)$ as the probability density of $x_t$ at any point in time $t$. The forward diffusion process can be modelled as the solution to the SDE

$$\mathrm{d}x_t = f(x_t, t)\,\mathrm{d}t + g(t)\,\mathrm{d}\mathbf{w}, \tag{1}$$

where $f(x_t, t) : \mathbb{R}^{d+1} \to \mathbb{R}^d$ and $g(t) : \mathbb{R} \to \mathbb{R}$ are the diffusion and drift coefficients respectively, $\mathrm{d}t$ represents an infinitesimal time step, and $\mathrm{d}\mathbf{w}$ is the differential of a standard Wiener process (Brownian motion).

If $f(x_t, t)$ is chosen to be an affine transformation of $x_t$, then the perturbation kernel of the SDE is a Gaussian distribution [48]. In this work, we set $f(x_t, t) := -\frac{1}{2}\beta(t)x_t$ and $g(t) := \sqrt{\beta(t)}$, following the *variance preserving* SDE [38]. This also corresponds the continuous generalisation of the Denoising Diffusion Probabilistic Model [44]. Here, $\beta(t)$ represents the strength of the Gaussian perturbation kernel at each stage of the diffusion process, giving

$$\mathrm{d}x_t = -\frac{1}{2}\beta(t)x_t\,\mathrm{d}t + \sqrt{\beta(t)}\,\mathrm{d}\mathbf{w}. \tag{2}$$

Typically, $\beta(t=0) = 0$ and is a monotonically increasing function.

New samples following $x(t = 0) \sim p_{\text{data}}$ can be generated by drawing from the prior and reversing the entire diffusion process. This relies on the fact that for any diffusion SDE of the form in Eq. (2), the reverse process is also an SDE [49] given by

$$\mathrm{d}\boldsymbol{x}_t = -\frac{1}{2}\beta(t)\big[\boldsymbol{x}_t + 2\nabla_{\boldsymbol{x}_t}\log p(\boldsymbol{x}_t)\big]\mathrm{d}t + \sqrt{\beta(t)}\,\mathrm{d}\bar{\mathbf{w}}, \tag{3}$$

where $\mathrm{d}\bar{\mathbf{w}}$ is the differential Wiener process when reversing the flow of time. The $\nabla_{\boldsymbol{x}_t}\log p(\boldsymbol{x}_t)$ term is referred to as the score function [47]. It is the gradient of the log-probability of the diffused data. The solution for the reverse SDE has the same marginal probabilities $p(\boldsymbol{x}_t)$ as the forward SDE. Alternatively, instead of generating new samples using the reverse SDE, there exists a deterministic ordinary differential equation (ODE) which preserves $p(\boldsymbol{x}_t)$. This is called the *probability-flow* ODE [38], and is given by

$$\mathrm{d}\boldsymbol{x}_t = -\frac{1}{2}\beta(t)\big[\boldsymbol{x}_t + \nabla_{\boldsymbol{x}_t}\log p(\boldsymbol{x}_t)\big]\mathrm{d}t. \tag{4}$$

For a given choice of $\beta(t)$, the only unknown term in either differential equation is the score function $\nabla_{\boldsymbol{x}_t}\log p(\boldsymbol{x}_t)$. If the score function can be determined, it is possible to generate samples under $p_{\text{data}}$ by first sampling from $p_{\text{prior}}$ and using either the reverse SDE or the *probability-flow* ODE in combination with either annealed Langevin dynamics [45], numerical SDE solvers [38,50], or numerical ODE solvers [39,51–53].

## 3.1 Modelling the score function

Although the score function may not be well-defined, it is possible to learn an approximation from data using a parametrised model known as a score-based model [45]. This is often a time conditional neural network $s_\theta(\boldsymbol{x}_t, t) \approx \nabla_{\boldsymbol{x}_t}\log p(\boldsymbol{x}_t)$ with parameters $\theta$, and it can be trained using the denoising score matching objective [54–56]. The training loss is defined by first decomposing the expectation over the perturbed data density $p(\boldsymbol{x}_t) = p(\boldsymbol{x}_0)p(\boldsymbol{x}_t|\boldsymbol{x}_0)$ and by conditioning the score function on the original data $\nabla_{\boldsymbol{x}_t}\log p(\boldsymbol{x}_t|\boldsymbol{x}_0)$. This conditional density can be defined through the Gaussian perturbation kernel $p(\boldsymbol{x}_t|\boldsymbol{x}_0) = \mathcal{N}(\boldsymbol{x}_t; \gamma(t)\boldsymbol{x}_0, \sigma(t)^2\boldsymbol{I})$, where $\gamma(t)$ and $\sigma(t)$ are referred to as the *signal rate* and the *noise rate* respectively, and are related to $\beta(t)$ by

$$\gamma(t) = \exp\left(-\frac{1}{2}\int_0^t \beta(s)\,\mathrm{d}s\right) \quad \text{and} \quad \sigma(t)^2 = 1 - \exp\left(-\int_0^t \beta(s)\,\mathrm{d}s\right). \tag{5}$$

As with Ref. [39,57], we choose to define $\gamma(t)$ and $\sigma(t)$ directly, and then derive a form for $\beta(t)$. For the signal and noise rates we use a variant of the cosine diffusion schedule from Ref. [57] but with a slight change in parameterization for the variance preserving SDE. We define a maximum and a minimum signal rate, $\sigma_{\max}$ and $\sigma_{\min}$, and use them to define the schedules with

$$\lambda_a = \arccos(\sigma_{\max}), \tag{6}$$

$$\lambda_b = \arccos(\sigma_{\min}), \tag{7}$$

$$\gamma(t) = \cos\big(\lambda_a + t(\lambda_b - \lambda_a)\big), \tag{8}$$

$$\sigma(t) = \sin\big(\lambda_a + t(\lambda_b - \lambda_a)\big). \tag{9}$$

We find the best results with $\sigma_{\max} = 0.999$ and $\sigma_{\min} = 0.02$. Since $\sigma_{\max} \approx 1$, we can use a simple approximation for $\beta(t)$, with

$$\beta(t) \approx 2(\lambda_b - \lambda_a)\tan\big(\lambda_a + t(\lambda_b - \lambda_a)\big). \tag{10}$$

We can sample from $p(\boldsymbol{x}_t|\boldsymbol{x}_0)$ using the re-parametrisation trick $\boldsymbol{x}_t = \gamma(t)\boldsymbol{x}_0 + \sigma(t)\boldsymbol{\epsilon}$, where $\boldsymbol{\epsilon} \sim \mathcal{N}(\boldsymbol{0}, \boldsymbol{I})$, and we can easily take the gradient of its logarithm leading to a tractable expression for the conditional score function $\nabla_{\boldsymbol{x}_t} \log p(\boldsymbol{x}_t|\boldsymbol{x}_0) = -\boldsymbol{\epsilon}/\sigma(t)$. Using this, the optimization problem becomes

$$\min_{\theta} \mathbb{E}_{t \sim \mathcal{U}(0,1)} \mathbb{E}_{\boldsymbol{x}_0 \sim p(\boldsymbol{x}_0)} \mathbb{E}_{\boldsymbol{\epsilon} \sim \mathcal{N}(\boldsymbol{0}, \boldsymbol{I})} \frac{1}{\sigma(t)^2} \|\hat{\boldsymbol{\epsilon}}_{\theta}(\boldsymbol{x}_t, t) - \boldsymbol{\epsilon}\|^2, \tag{11}$$

where $\hat{\boldsymbol{\epsilon}}_{\theta}(\boldsymbol{x}_t, t) = -\sigma(t)\boldsymbol{s}_{\theta}(\boldsymbol{x}_t, t)$ is a re-parametrization of the score-based model.

One major drawback of this optimisation process is that the $\sigma(t)^2$ in the denominator causes the loss to explode as $\sigma(t) \to 0$. Loss reweighing schemes [39] introduce a positive scalar weight $\lambda(t)$ in front of the training objective and have been found to improve the quality of generation. Setting $\lambda(t) = \sigma(t)^2$ cancels out the term in the denominator, and the increased stabilisation leads to good perceptual quality in image generation. On the other hand, setting $\lambda(t) = \beta(t)$ corresponds to training the model to maximize the log-likelihood of the data through the negative evidence lower bound, but training still suffers from instability issues. Our training objective uses a sum of these two terms with a relative weight hyperparameter $\alpha$ [57].

Extending score-based models to conditional generative models can be performed by providing some conditional vector $\boldsymbol{y}$ to the score-based model and sampling from the joint distribution of the data during training. Our final training objective is therefore given by

$$\min_{\theta} \mathbb{E}_{t \sim \mathcal{U}(0,1)} \mathbb{E}_{\boldsymbol{x}_0, \boldsymbol{y} \sim p(\boldsymbol{x}_0, \boldsymbol{y})} \mathbb{E}_{\boldsymbol{\epsilon} \sim \mathcal{N}(\boldsymbol{0}, \boldsymbol{I})} \left(1 + \alpha \frac{\beta(t)^2}{\sigma(t)^2}\right) \|\hat{\boldsymbol{\epsilon}}_{\theta}(\boldsymbol{x}_t, t, \boldsymbol{y}) - \boldsymbol{\epsilon}\|^2. \tag{12}$$

In practice we find that using Huber loss instead of the Frobenius norm results in faster training and better generation quality. We performed a scan over $\alpha$ from $10^{-4}$ to $10^{-2}$ and found the best generative performance with $\alpha = 10^{-3}$.

Thus, the neural network we use for the score function $\hat{\boldsymbol{\epsilon}}_{\theta}(\boldsymbol{x}_t, t, \boldsymbol{y})$ is trained to predict the noise that has been introduced to produce the diffused data $\boldsymbol{x}_t$. This prediction is calculated for a given time $t \in [0, 1]$ in the diffusion process and additional contextual information $\boldsymbol{y}$ relating to the jet.

## 4 Generating jets with diffusion

In high energy collisions of protons at colliders such as the LHC, quarks and gluons (partons) are produced in large quantities and from a wide range of processes. Partons cannot exist as free states due to colour confinement, and instead radiate other partons before hadronising into a shower of colour neutral hadrons. These collimated showers of hadrons subsequently interact with the detector material, leaving signatures of electrically charged and neutral particles within a cone originating from the interaction point. After a clustering process, these showers are reconstructed into single objects called jets. Due to the high multiplicity of particles, the complex and stochastic nature of the development of the shower, and the subsequent interaction with detector material, jets are computationally expensive to simulate.

At the LHC, quarks can be produced in the decays of particles such as $W/Z$ bosons, or top quarks through their decay into a $W$ boson and a $b$-quark. For the majority of energy scales the two or three quarks from these decays produce jets which can be individually resolved in the detector. However, as the momenta of the intermediate particles increase, the decay products themselves start to collimate resulting in a single large-radius jet in the detector (the so-called boosted regime). The vast majority of jets, however, are initiated by partons which are not the decay products of other massive particles (QCD background).

At the ATLAS and CMS experiments, constituents of jets are reconstructed from the trajectories of charged particles and energy deposits in dedicated calorimeter systems using the Particle Flow [58] algorithm; they are each represented by a four-momentum vector. These constituents are clustered into jets using the anti-$k_t$ algorithm [59]. Typically, constituents are clustered with a radius parameter $R = 0.4$ into small-radius (resolved) jets. However, in this work we only consider jets in the boosted regime; a radius parameter of $R = 0.8$ is used.[1] The four-momentum vector of a jet is calculated from the vector sum of its constituents.

## 4.1 Jet substructure

Properties relating to the distribution of constituents within a jet are known as the substructure of a jet, which can be used to identify the original seed particle [60]. This is particularly interesting in the boosted regime, where several partons from the decay of another elementary particle have overlapping showers.

One commonly used set of observables to describe jet substructure is its *N-subjettiness* [61], denoted by $\tau_N$. These are useful in identifying jets originating from processes with $N$ prongs as a result of the decay of the initial particle. A jet originating from a gluon is likely to have a 1-prong substructure, whereas a $W/Z$ boson decay is likely to produce a 2-prong jet, and a jet originating from the all-hadronic decay of a top quark will tend to be 3-prong. Other commonly used observables relate to the energy correlation functions of a jet and their ratios, such as $D_2$ [62, 63].[2] A new set of features which have been found to be sensitive to the underlying substructure of different jet types are the Energy Flow Polynomials (EFPs) [64].

Furthermore, when a seed particle decays, the observed opening angle of the decay products is strongly dependent on its mass and momentum. In jets, this means that the distribution of constituent properties is strongly correlated to the overall invariant mass and transverse momentum $p_T$.

Classification approaches applying cuts on the substructure features, as well as machine learning algorithms trained using such features have been successfully employed in the ATLAS and CMS collaborations to distinguish jets originating from $W$ bosons ($W$-jets), top quarks (top jets), gluons, and light quarks [65, 66]. In recent years, more sophisticated classification algorithms have been trained on the constituents themselves, either represented as ordered vectors [65–68], images [69–71], or point clouds [72–80] (see Ref. [4] for a review).

These approaches are very sensitive to the substructure of jets originating from different particles. As such, when using fast surrogate models it is crucial that they accurately capture the distribution of the constituents within a jet and their correlations to the mass and $p_T$.

## 4.2 Datasets

In this work we focus on the generation of two classes of jets defined by the particle they originate from, gluons and top quarks. Gluon-initiated jets are the dominant background in proton-proton colliders, while boosted top quarks produce jets with rich substructures due to the nature of their decay. These jet types provide key benchmark datasets to probe the behaviour of the model, and enable comparisons with other approaches.

For these studies we use the JetNet30 datasets [81] provided by the JetNet v0.2.2 package introduced in Ref. [33], the same dataset used to train MPGAN. These datasets consist of large-radius jets simulated in a generic detector at a proton-proton collider with a centre of mass energy $\sqrt{s} = 13$ TeV. They are selected to have transverse momenta of approximately 1 TeV. Each jet is described by its 30 leading $p_T$ constituents, which are themselves described by their

---

[1]This corresponds to the radius parameter used by the CMS collaboration.

[2]$D_2$ is defined as the ratio of the three-point and cubed two-point energy correlation functions.

three momentum vectors with coordinates relative to the jet ($\Delta\eta$, $\Delta\phi$, $p_{\mathrm{T}}$).[3] The relative pseudorapidity is defined as $\Delta\eta = \eta^{\mathrm{const.}} - \eta^{\mathrm{jet}}$, and the relative azimuthal angle is defined as $\Delta\phi = \phi^{\mathrm{const.}} - \phi^{\mathrm{jet}}$.

For conditional generation, the combined mass and transverse momentum of the point cloud ($p_{\mathrm{T}}^{\mathrm{jet}}$, $m_{\mathrm{jet}}$) are provided during training and generation. These observables are calculated from the four-momentum vectors of the selected jet constituents. As this dataset only uses a subset of the original jet constituents, the true transverse momentum and invariant mass of jets with more than 30 constituents are greater than $p_{\mathrm{T}}^{\mathrm{jet}}$ and $m_{\mathrm{jet}}$.[4] For a fair comparison to MPGAN we use the same train and test splits chosen in Ref. [33].

## 4.3  PC-JeDi architecture and training

PC-JeDi is a conditional score-based diffusion model trained to predict the noise added to a diffused particle cloud given two conditional properties of the jet, $p_{\mathrm{T}}^{\mathrm{jet}}$ and $m_{\mathrm{jet}}$. The choice of neural network for the score model is open, though a desirable property for the set-to-set mapping is permutation equivariance. A wide variety of appropriate neural network architectures can be used for PC-JeDi including graph neural networks [82] and deep sets [83]. For these studies, we use attention based transformers [84] which are an efficient and expressive class of neural networks based on the self-attention mechanism. Their operations are equivariant under the permutation of the input tokens, which here are the jet constituents represented by their kinematics.

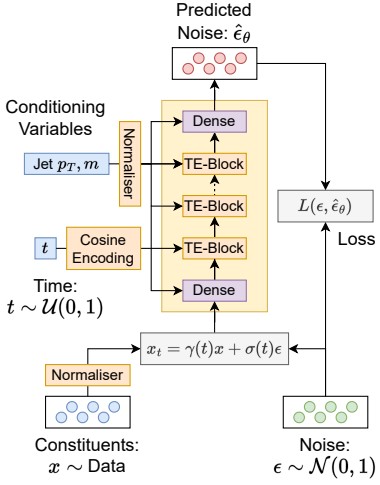
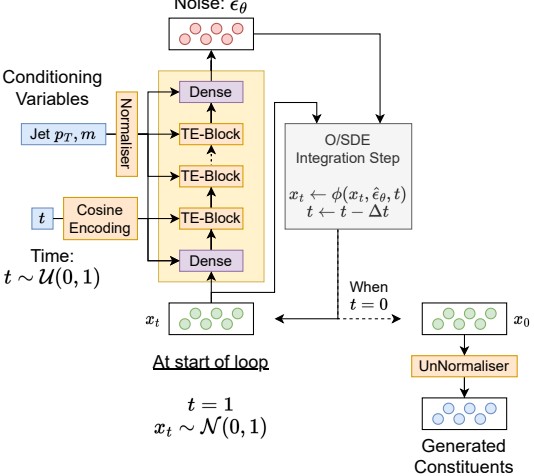

Figure 1: Diagram of the PC-JeDi training procedure. Data and noise are mixed together according to the signal and noise schedulers. Then the conditioned model is optimised via a distance loss between the noise it predicts and the original noise.

Figure 2: Diagram of the PC-JeDi generation procedure. Random noise is sampled at the beginning of the loop from a standard normal distribution. Then any chosen integration sampler is iteratively applied in order to fully denoise the input towards actual data, using the conditional model as a noise predictor.

The model receives three inputs: the set of noise augmented constituents $\boldsymbol{x}_t$, the diffusion time parameter $t$, and the conditional variables of the jet $\boldsymbol{y} := (p_{\mathrm{T}}^{\mathrm{jet}}, m_{\mathrm{jet}})$. The constituents are represented by their relative coordinates and absolute $p_{\mathrm{T}}$ values $\boldsymbol{x}_t := (\Delta\eta, \Delta\phi, \log(p_{\mathrm{T}} + 1))$.

---

[3]Jets with fewer than 30 constituents are zero-padded and a binary mask is provided in order to identify them.
[4]For application as a fast surrogate model, the $p_{\mathrm{T}}$ and $m_{\mathrm{jet}}$ calculated from all constituents would be used, and the model would not be restricted to 30 constituents.

The constituent $p_T$ is transformed with a log operation to improve reconstruction of low momenta constituents.

The model is built using several chained Transformer-Encoder (TE) blocks [84], each employing multi-headed self-attention. An initial dense network is used to embed the input point cloud into a larger space in order for the self-attention mechanism to be sufficiently expressive. The final dense network reshapes the output tokens back to the original input dimension. To ensure that the dynamic range of the data is kept within reasonable bounds, the conditional and constituent variables are passed through normalisation layers, which shift and rescale each variable to zero mean and unit variance, with values calculated from the training dataset. The diffusion time parameter is passed through a cosine encoding layer, producing an $M$-dimensional vector of increasing frequencies $\boldsymbol{v}_t = \left(\cos(e^0 t\pi), \cos(e^1 t\pi), ..., \cos(e^{M-1} t\pi)\right)$.

Figure 1 shows the model and information flow during the training procedure. During training, the network learns to predict how much noise has been used to perturb an input jet. First, jet constituents in the form of a set $\boldsymbol{x}$ and the corresponding jet features $\boldsymbol{y}$ are sampled from the data. For the noise, an equal sized set of points $\boldsymbol{\epsilon}$ are sampled from a standard normal distribution, and a diffusion time $t \sim \mathcal{U}(0,1)$ is sampled from a uniform distribution. To get the perturbed input $\boldsymbol{x}_t = \gamma(t)\boldsymbol{x} + \sigma(t)\boldsymbol{\epsilon}$, a weighted sum of the jet constituents and the noise is computed using the time-dependent signal and noise scales, $\gamma(t)$ and $\sigma(t)$. This perturbed point cloud is passed to the network along with the conditional information and the encoded time vector to get a prediction of the initial noise $\hat{\boldsymbol{\epsilon}}_\theta$. A distance measure between this prediction noise and the true noise is used as the objective function to train the network.

To generate new sets of jet constituents, the trained network is used to iteratively denoise a point cloud that has been sampled from a standard normal distribution. This procedure is shown in Fig. 2. First, the diffusion time is set to $t = 1$, the input point cloud is sampled from a standard normal distribution $\boldsymbol{x}_{t=1} \sim \mathcal{N}(0,1)$, and the desired jet properties $\boldsymbol{y}$ are chosen. As before, the model attempts to predict the noise component of $\boldsymbol{x}_t$. This output is used to model the score-function of the data, which is needed by the integration solver to update $\boldsymbol{x}_t$ for a small negative time step $\Delta t$. The whole procedure is repeated until the diffusion time reaches $t = 0$. The resulting output of the integration sampler $\boldsymbol{x}_{t=0}$ corresponds the fully generated set of constituents, given the chosen jet features. Both the training and generation procedures are summarised in Appendix A.

It is worth noting that the network, the training procedure, and choice of integration method are independent pieces of the whole implementation. This leads to several advantages. First, the solver can be selected after the training procedure is completed. This allows for some level of optimisation and fine-tuning of the method without retraining the network. Furthermore, as new solvers are developed, the existing trained network can still be used. Second, the score-based training method can be used to train any network architecture for predicting the noise, not just the transformer we present here.

In PC-JeDi separate networks are trained to generate either top quark or gluon jets for the chosen transverse momentum and invariant mass. The hyperparameters for the model and training setup are discussed in Appendix B, and the PC-JeDi source code is publicly available.[5]

## 4.4 Evaluation metrics

To evaluate the performance of PC-JeDi we use the same set of measures as introduced in Ref. [33]. These include the distribution over reconstructed jet masses, the inclusive marginals over constituent four momenta, and the average values of a subset of EFPs. As an extension to these measures, we also look at the leading, sub-leading and sub-sub-leading constituent four momenta of each jet, ordered in decreasing $p_T$.

---

[5]https://github.com/rodem-hep/PC-JeDi

In addition to the observables studied in Ref. [33], we also look at the jet $N$-subjettiness ratios $\tau_{32}$ and $\tau_{21}$ distributions ($\tau_{ij} = \tau_i/\tau_j$) and the energy correlation function ratio $D_2$ distributions. These observables are commonly used for the identification of top jets, and they are strongly linked to the invariance mass and transverse momentum of the jets. We look at the two-dimensional marginals of these distributions and the invariant mass of the jet in order to observe whether the underlying correlations are correctly modelled.

To enable comparison to jets generated with MPGAN, we calculate observables using the relative $p_T$ of constituents with respect to the jet. It is defined as $p_T/p_T^{\text{jet}}$ per constituent. We denote kinematic and substructure observables calculated using $p_T^{\text{rel}}$ in place of $p_T$ with the superscript 'rel'. This alters the scale of the resulting observable but tests the same underlying physics. For the $N$-subjettinness ratios, the scale effects cancel.

As a quantitative measure we calculate the distributional distances between the MC and generated jets using the averaged Wasserstein-1 distance metric $W_1$. $W_1^M$ is the distance between the distributions of the constituents relative mass $m_j^{\text{rel}}$, $W_1^P$ is the average distance between the distributions of the constituents three momentum ($\Delta\eta, \Delta\phi, p_T^{\text{rel}}$), and $W_1^{\text{EFP}}$ is the average $W_1$ using the first five EFPs. The distances of the $N$-subjettiness ratios and $D_2$ are denoted by $W_1^{\tau_{32}}$, $W_1^{\tau_{21}}$ and $W_1^{D_2}$.

On top of the Wasserstein-1 distance of the distributions, we compute the Fréchet ParticleNet distance (FPND) [33] which compares the mean and standard deviation of the penultimate layer of the ParticleNet model for the MC and generated jets [33, 72]. We also look at the coverage (Cov) and the minimum matching distance (MMD) metrics as described in Ref. [33].

## 5 Results

In order to generate jets with PC-JeDi it is first necessary to choose an integration sampler for using in the generation procedure. The approach of formulating diffusion models as SDEs/ODEs is still in active development, and there is no clear consensus on the best method to use. To cover a broad range, we study one approach to solve the SDE in Eq. (3) and two different methods to solve the ODE in Eq. (4). Additionally, we investigate the impact of a solver designed specifically for generative diffusion models. However, these do not form an exhaustive comparison. All approaches use the same trained network.

The Euler-Maruyama (EM) algorithm [85] is used for integrating the SDE, which yields the exact solution to the reverse SDE. To solve the *probability-flow* ODE, we examine two solvers: the standard Euler solver and the fourth-order Runge-Kutta (RK) method [53]. The RK method is an extension of the Euler method, which considers multiple values of the integrated function within the integration interval. It emphasises the midpoint value rather than the edges of the integration step. Finally, we evaluate the DDIM solver [51]. This is a deterministic solver specifically designed for diffusion generative models. It predicts $\boldsymbol{x}_0$ directly at each stage of the reverse process and uses it to define the update. The detailed algorithms for these four integration samplers are provided in Appendix C.1.

In choosing a solver there is also a trade-off between the quality of the generated samples and the generation speed. This arises in optimising the number of integration steps. Performing the integration over more steps requires more forward passes through the network. This should result in higher quality generated jets, but increases the required generation time. This trade-off is studied in Appendix C.2.

In the following results, we choose to focus on generation quality rather than speed of generation. All jets are generated using 200 integration steps and we focus on the DDIM and EM solvers. Negligible improvement in quality is observed beyond this number of steps, and

at this point difference between solvers is small. A comparison of all solvers and additional observables can be found in Appendix D.

## 5.1  Inclusive generation of jets

First we focus on the inclusive generation of jets following the same $p_T^{jet}$ and $m_{jet}$ distributions seen during training. This allows us to compare directly to the non-conditional MPGAN model.[6] For these comparisons we calculate the relative transverse momentum ($p_T^{rel}$) of each constituent using the full $p_T$ of the jet.

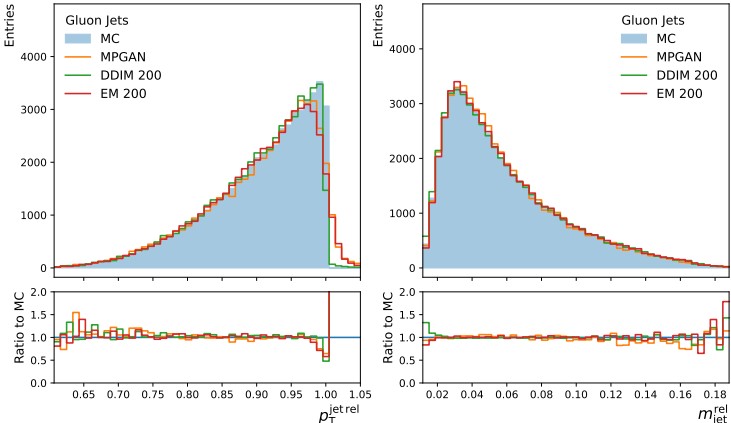

Figure 3: The relative transverse momentum (left) and invariant mass (right) of gluon jets generated with MPGAN (orange) and PC-JeDi (DDIM solver, green; EM solver, red) compared to the MC simulation (shaded blue). Calculated from the leading 30 $p_T$ constituents using the constituent $p_T^{rel}$ instead of $p_T$.

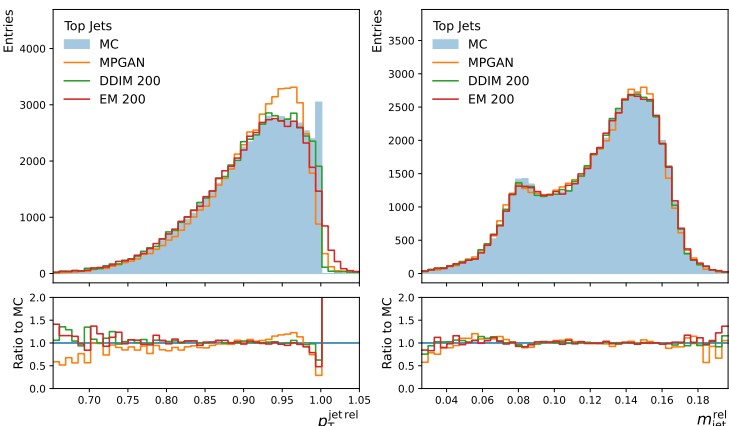

Figure 4: The relative transverse momentum (left) and invariant mass (right) of top jets generated with MPGAN (orange) and PC-JeDi (DDIM solver, green; EM solver, red) compared to the MC simulation (shaded blue). Calculated from the leading 30 $p_T$ constituents using the constituent $p_T^{rel}$ instead of $p_T$.

We look at the relative transverse momentum $p_T^{rel}$ and relative invariant mass $m_{jet}^{rel}$ of the reconstructed jet. These observables are calculated from the vector sum of the 30 (leading in $p_T$) jet constituents using $p_T^{rel}$ in place of $p_T$ per constituent. These are shown in Figs. 3

---

[6]For fair comparisons we use the trained model provided by Ref. [33] and generate an equal number of jets with both PC-JeDi and MPGAN and evaluate all metrics consistently for both models using the `JetNet` library provided with the datasets.

and 4 for gluon jets and top jets, respectively. The value of $p_T^{\text{jet, rel}}$ is not always exactly 1.0 due to selecting only the leading 30 constituents for each jet. However this is the maximum physical value. All generative models struggle to capture the hard cut off at 1.0 in $p_T^{\text{jet, rel}}$, though PC-JeDi with the DDIM solver is closest in agreement. Both PC-JeDi models outperform MPGAN at reconstructing the top jet $p_T^{\text{rel}}$ distribution. All three models perform similarly at reproducing the $m_{\text{jet}}^{\text{rel}}$ for both top quarks and gluons.

It is also important that the individual constituents are accurately modelled. In Figs. 5 and 6 we see that the relative transverse momentum of the leading three constituents ordered by $p_T$ are well reproduced by MPGAN and PC-JeDi for both gluon and top jets. However, both PC-JeDi models show disagreements at low values of transverse momentum for gluon jets. Here, MPGAN is better able to capture these values. The relative $\eta$ and $\phi$ coordinates of the constituents are found to be in good agreement with the MC simulation for all three models.

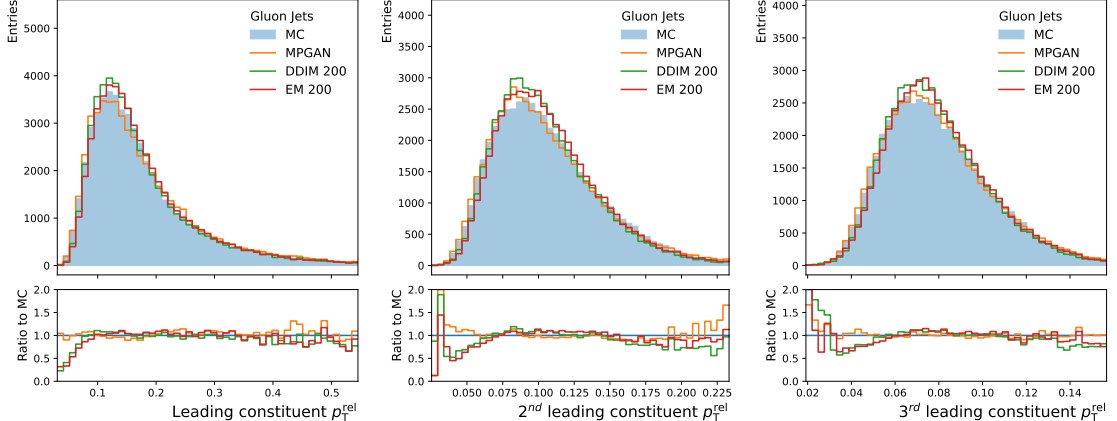

Figure 5: Distributions of the leading (left), subleading (middle) and third leading (right) constituent $p_T^{\text{rel}}$ for the gluon jets generated with MPGAN (orange) and PC-JeDi (DDIM solver, green; EM solver, red) compared to the MC simulation.

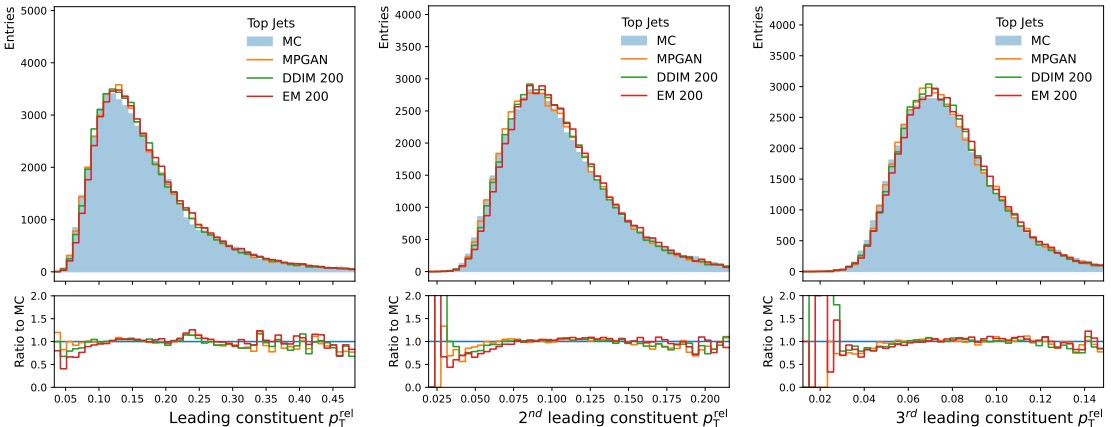

Figure 6: Distributions of the leading (left), subleading (middle) and third leading (right) constituent $p_T^{\text{rel}}$ for the top jets generated with MPGAN (orange) and PC-JeDi (DDIM solver, green; EM solver, red) compared to the MC simulation.

Finally, we look at the relative $\tau_{21}$, $\tau_{32}$ and $D_2$ substructure observables in Figs. 7 and 8. Both PC-JeDi and MPGAN are able to capture the $D_2$ distributions, with MPGAN visually showing better agreement. However all three models struggle to capture both $\tau_{21}$ and $\tau_{32}$ for gluon jets. This is even more apparent for top jets, which have a bi-modal structure in all three observables.

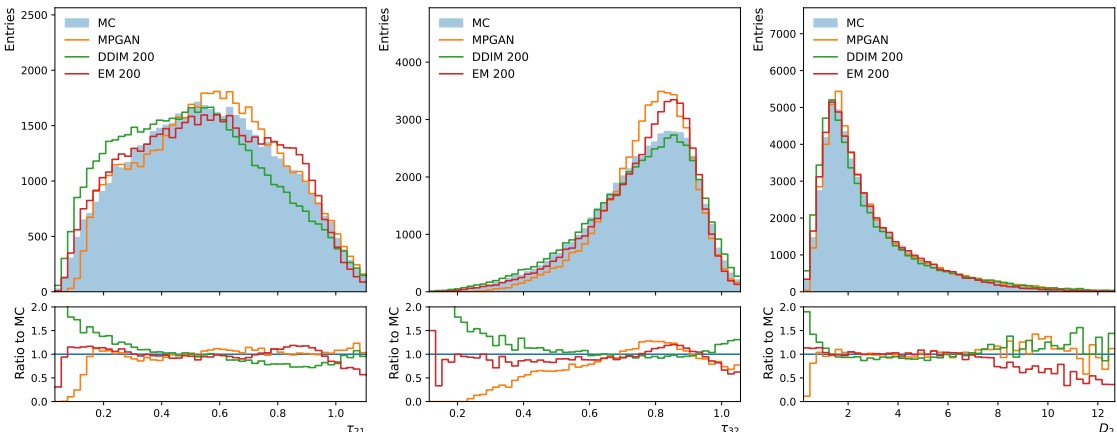

Figure 7: Relative substructure distributions $\tau_{21}^{\text{rel}}$ (left), $\tau_{32}^{\text{rel}}$ (middle) and $D_2^{\text{rel}}$ (right) for gluon jets generated with MPGAN (orange) and PC-JeDi (DDIM solver, green; EM solver, red) compared to the MC simulation.

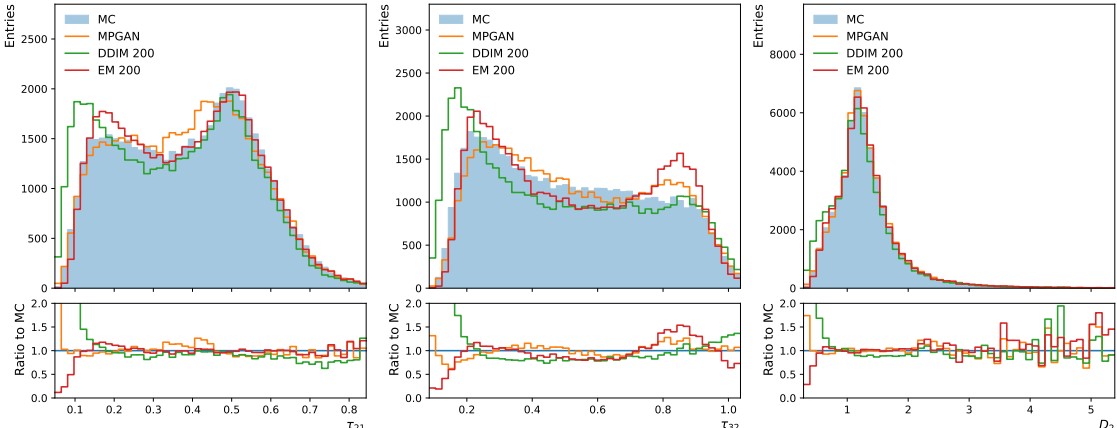

Figure 8: Relative substructure distributions $\tau_{21}^{\text{rel}}$ (left), $\tau_{32}^{\text{rel}}$ (middle) and $D_2^{\text{rel}}$ (right) for top jets generated with MPGAN (orange) and PC-JeDi (DDIM solver, green; EM solver, red) compared to the MC simulation.

The performance is compared quantitatively in Table 1 for the metrics introduced in Ref. [33] and Table 2 for the additional substructure distributions in Figs. 7 and 8. For each metric we establish an ideal limit by comparing the training and test sets, which corresponds to the natural variation in the MC samples. Following the procedure defined by Ref. [33] uncertainties for the Wasserstein based metrics are derived using bootstrap sampling, however we increase the number of bootstrapped batches from 5 to 40 to reduce the run to run variance. The FPND metric requires the entire test set and does not use bootsrapping so we do not quote an uncertainty. PC-JeDi beats the current methods for several metrics and is competitive across several others for both gluon and top jet generation. For top generation PC-JeDi has a notably lower FPND and $W_1^P$ scores than MPGAN yet performs worse in $W_1^M$ and $W_1^{EFP}$. The metrics **Cov** and **MMD** are essentially saturated by all models, as they are in agreement with the upper limit defined by the natural variation in the MC samples. Some of the values seem to be in tension with visual inspection. Most notably, $W_1^P$ for gluon jets suggests PC-JeDi with the EM solver outperforms MPGAN, despite the observed underestimation at low values of $p_T^{rel}$ for the three leading constituent. This shows the importance of studying a wide range of distributions, and highlights a potential limitation in using aggregated $W_1$ distances. It may also suggest that a metric more sensitive to the behaviour in tails of distributions could be beneficial, for example classifier-based weight approaches [86].

Table 1: Comparison of metrics introduced in Ref. [33] for the generated jets. Lower is better for all metrics except Cov.

| Jet Class | Model | Sampler (steps) | FPND | $W_1^P(\times 10^{-3})$ | $W_1^{EFP}(\times 10^{-5})$ | $W_1^M(\times 10^{-3})$ | Cov ↑ | MMD ($\times 10^{-2}$) |
|---|---|---|---|---|---|---|---|---|
| Top | MC | - | 0.01 | 0.40 ± 0.13 | 0.81 ± 0.35 | 0.32 ± 0.11 | 0.59 ± 0.02 | 7.13 ± 0.24 |
| | MPGAN | - | 0.36 | 2.17 ± 0.20 | **1.28 ± 0.49** | **0.64 ± 0.21** | 0.58 ± 0.02 | **7.11 ± 0.13** |
| | PC-JeDi | DDIM (200) | 0.28 | **1.01 ± 0.11** | 4.12 ± 0.56 | 1.48 ± 0.31 | **0.59 ± 0.15** | 7.13 ± 0.23 |
| | | EM (200) | **0.15** | 1.21 ± 0.20 | 3.56 ± 0.49 | 1.36 ± 0.32 | **0.59 ± 0.18** | **7.11 ± 0.22** |
| Gluon | MC | - | 0.01 | 0.41 ± 0.13 | 0.35 ± 0.12 | 0.44 ± 0.16 | 0.55 ± 0.03 | 3.67 ± 0.22 |
| | MPGAN | - | **0.13** | 1.03 ± 0.15 | 0.88 ± 0.24 | 0.82 ± 0.21 | 0.53 ± 0.03 | **3.60 ± 0.30** |
| | PC-JeDi | DDIM (200) | 0.12 | 0.66 ± 0.25 | **0.50 ± 0.09** | 0.90 ± 0.18 | **0.54 ± 0.02** | 3.64 ± 0.27 |
| | | EM (200) | 0.10 | **0.58 ± 0.14** | 0.57 ± 0.11 | **0.57 ± 0.15** | 0.54 ± 0.01 | 3.62 ± 0.19 |

Table 2: Wasserstein-1 distances for substructure observables between the generated jets and MC simulation, $W_1^{\tau_{21}}$, $W_1^{\tau_{32}}$, and $W_1^{D_2}$, and the mean absolute error between the reconstructed jet mass and transverse momentum and the target conditions $MAE^M$ and $MAE^{p_T}$. Lower is better for all metrics.

| Jet Class | Model | Sampler (steps) | $W_1^{\tau_{21}}(\times 10^{-3})$ | $W_1^{\tau_{32}}(\times 10^{-3})$ | $W_1^{D_2}(\times 10^{-2})$ | $MAE^M(\times 10^{-2})$ | $MAE^{p_T}(\times 10^{-2})$ |
|---|---|---|---|---|---|---|---|
| Top | MC | - | 2.01 ± 0.74 | 2.90 ± 1.59 | 1.23 ± 0.23 | - | - |
| | MPGAN | - | 6.61 ± 0.92 | 17.41 ± 2.78 | 3.30 ± 0.50 | - | - |
| | PC-JeDi | DDIM (200) | **4.40 ± 1.03** | 32.04 ± 2.29 | 2.59 ± 0.41 | **0.06** | **0.44** |
| | | EM (200) | 4.55 ± 1.16 | **16.05 ± 1.31** | **2.10 ± 0.43** | 0.19 | 1.24 |
| Gluon | MC | - | 3.79 ± 1.42 | 2.26 ± 0.51 | 3.93 ± 0.15 | - | - |
| | MPGAN | - | 16.83 ± 2.08 | 25.27 ± 1.29 | **6.08 ± 0.90** | - | - |
| | PC-JeDi | DDIM (200) | **11.99 ± 1.12** | 20.38 ± 1.91 | 11.39 ± 1.42 | **0.05** | **0.44** |
| | | EM (200) | 12.48 ± 0.98 | **13.32 ± 0.96** | 10.20 ± 1.04 | 0.10 | 1.29 |

Capturing the correlations between jet substructure observables and the jet kinematics is also a key measure of performance. Cuts on $\tau_{32}$ and $D_2$ are applied to distinguish top jets from gluon or quark jets in simple cut-based analyses [65, 66], with cut values are often derived as a function of the jet mass. Similarly $\tau_{21}$ is important in $W$-jet identification. Figures 9 and 10 show the distributions of these observables alongside the two-dimensional marginals for PC-JeDi with the EM solver and MPGAN. For gluon jets, PC-JeDi captures the correlations between features better than MPGAN. For top jets, both MPGAN and PC-JeDi capture the bimodal structure of the top jets with MPGAN showing slightly better agreement.

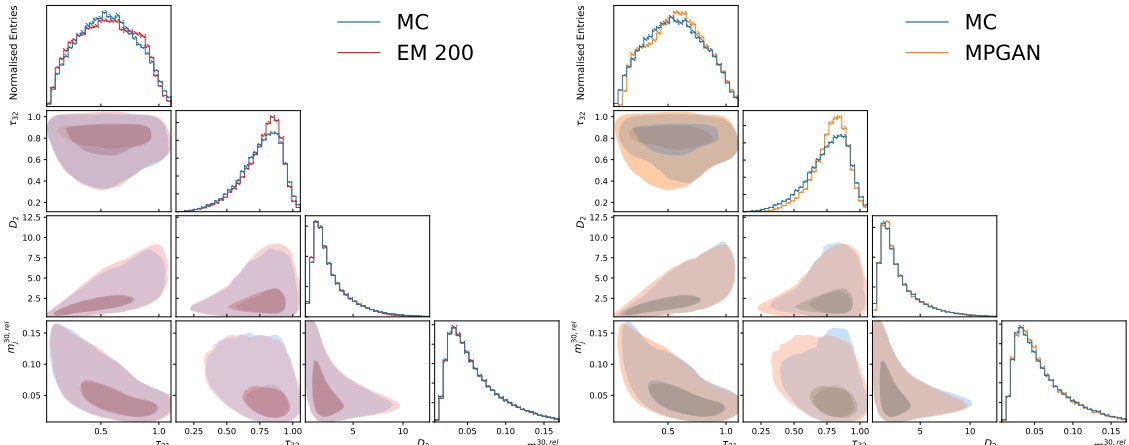

Figure 9: Mass and relative substructure distributions of the generated gluon jets using the EM solver for PC-JeDi, and MPGAN. The diagonal consists of the marginals of the distributions. The off-diagonal elements are the joint distributions of the variables.

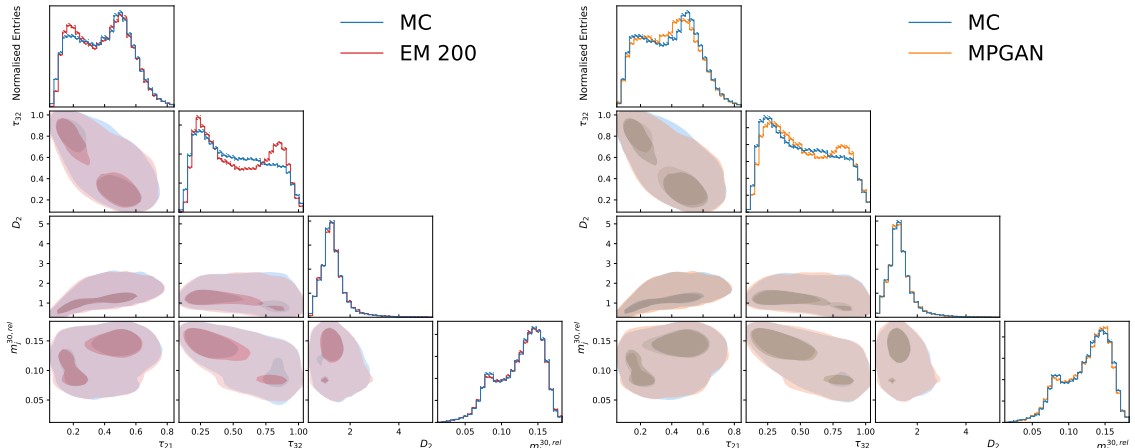

Figure 10: Mass and relative substructure distributions of the generated top jets using the EM solver for PC-JeDi, and MPGAN. The diagonal consists of the marginals of the distributions. The off-diagonal elements are the joint distributions of the variables.

## 5.2 Uncontained top jets

The bi-modal structure observed in top jets arises from a phenomenon in boosted top jet reconstruction where the stable particles from the $b$-quark decay are not contained within the radius of the jet. These top jets are referred to as *uncontained* top jets, and they exhibit a 2-pronged structure and masses close to the mass of the $W$ boson. This subset of jets is most visible in the inclusive jet mass distribution in Fig. 4, which shows a notable two peak structure corresponding to resonant $W$ decay and the full top jet decay.

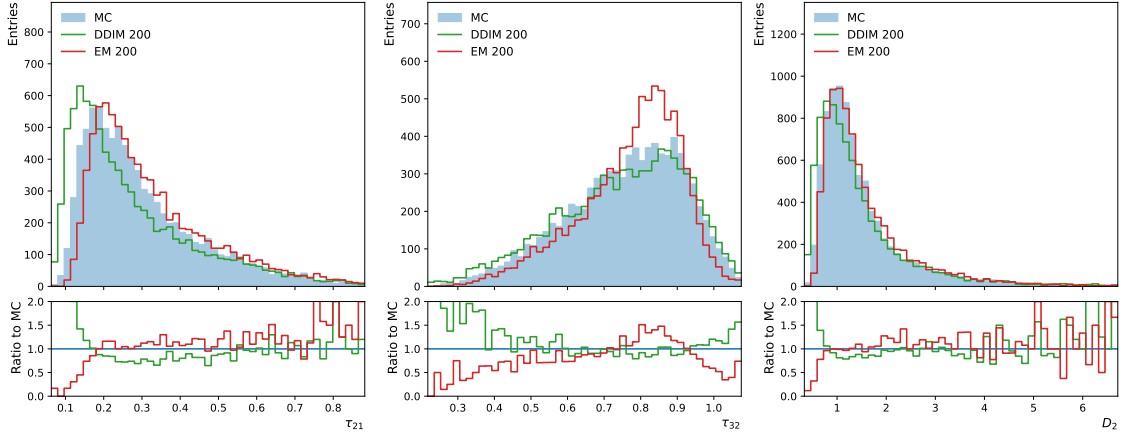

Figure 11: Relative substructure distributions $\tau_{21}$ (left), $\tau_{32}$ (middle) and $D_2$ (right) for uncontained top jets ($m_j \in [60, 100]$ GeV) generated with PC-JeDi (DDIM solver, green; EM solver, red) compared to the MC simulation.

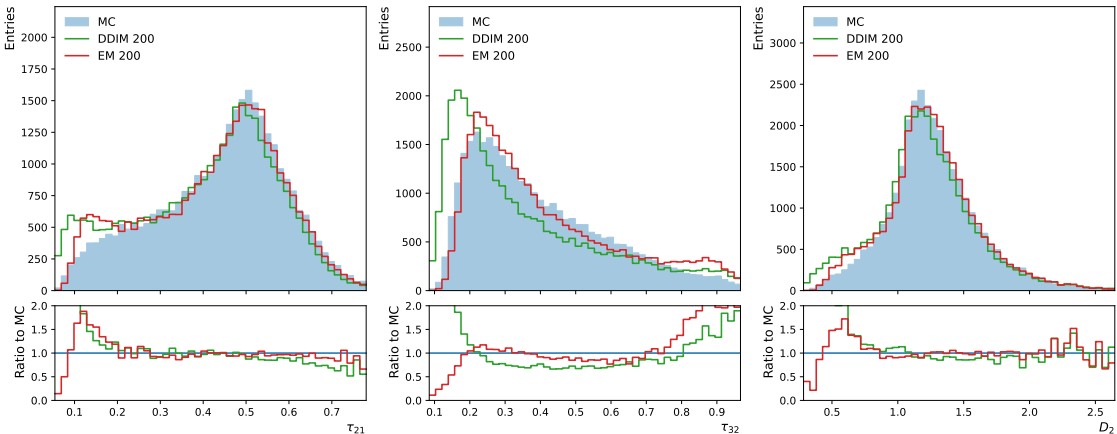

Figure 12: Relative substructure distributions $\tau_{21}$ (left), $\tau_{32}$ (middle) and $D_2$ (right) for contained top jets ($m_j \in [140, 200]$ GeV) generated with PC-JeDi (DDIM solver, green; EM solver, red) compared to the MC simulation.

In Figs. 11 and 12, we look at the distributions of $\tau_{21}$ and $\tau_{32}$ of jets generated with PC-JeDi in two mass windows. The first window ($m_j \in [60, 100]$ GeV) corresponds to the $W$ boson mass peak, in order to select uncontained top jets, and the second ($m_j \in [140, 200]$ GeV) is at the top quark mass peak to select fully contained top jets.

The EM solver reproduces the substructure distribution of the two populations fairly well in the bulk. However, for substructure variables which are strongly dependent on the soft constituent dynamics, such as $\tau_{32}$ and $\tau_{21}$, there are regions of phase space that deviate from the nominal. We also see that the DDIM solver performs generally better at these observables for uncontained top jets, whereas the opposite trend is true for EM. This demonstrates the

difficulty in choosing the optimal solver, with some better suited to different areas of phase space.

## 5.3 Conditional generation

As PC-JeDi is a conditional generation model, it is important to verify that the generated jets match the target transverse momentum and invariant mass. In Figs. 13 to 16 we compare the target and generated $m_{jet}^{30}$ and $p_T^{30}$ for gluon jets and top jets. In all cases, the DDIM solver shows a more linear correspondence between target and generation. However, in Figs. 15 and 16 we see that the DDIM solver results in off diagonal artefacts following diagonal lines for the $p_T^{30}$ distribution of both top jets and gluon jets. Nevertheless, these represent a much smaller fraction of events than the spread observed for EM in the same figures, and is at most 1% of the total number of generated jets for either solver. Furthermore, in Fig. 14 we see that both the DDIM and EM solvers exhibit an off diagonal spread in the generated top jet mass for target values corresponding to the $W$ mass peak of uncontained top jets.

We quantify the performance at conditional generation using the mean absolute error (MAE) between the generated and conditional jet mass and $p_T$ values in Table 2.

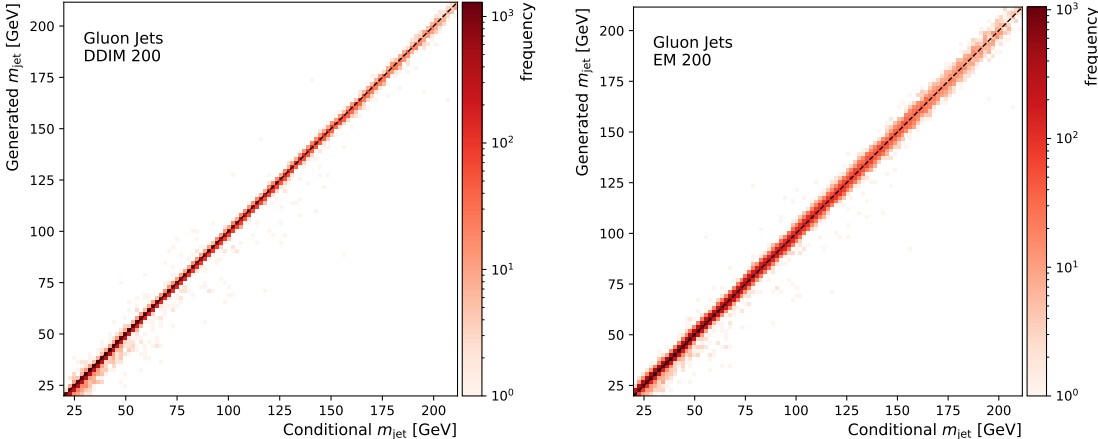

Figure 13: Two-dimensional histograms showing the correlation between the conditional and generated jet mass for the gluon jets using DDIM and EM solvers.

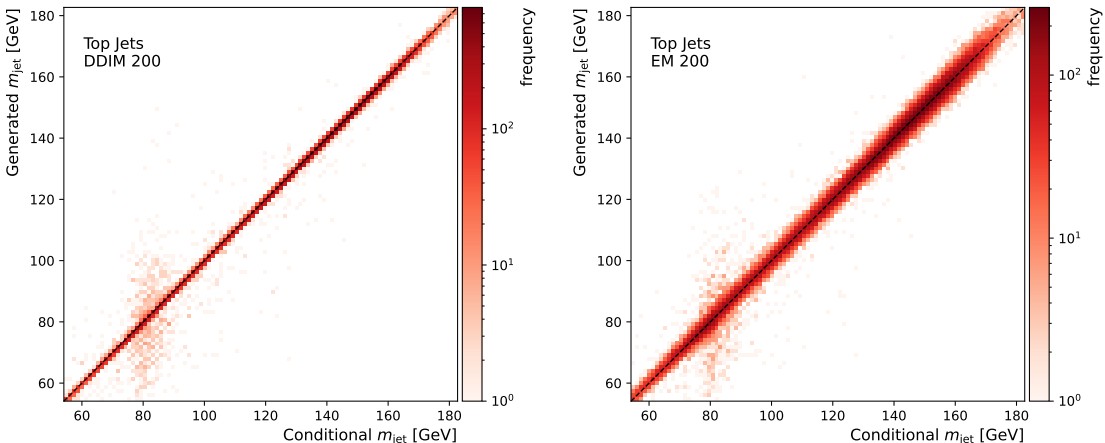

Figure 14: Two-dimensional histograms showing the correlation between the conditional and generated jet mass for the top jets using DDIM and EM solvers.

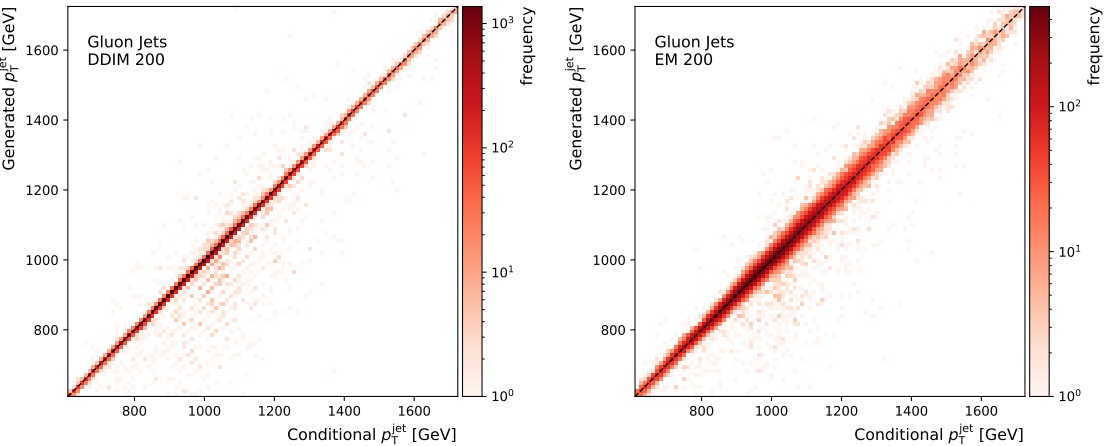

Figure 15: Two-dimensional histograms showing the correlation between the generated and conditional jet mass for the top jets using the DDIM and EM solvers.

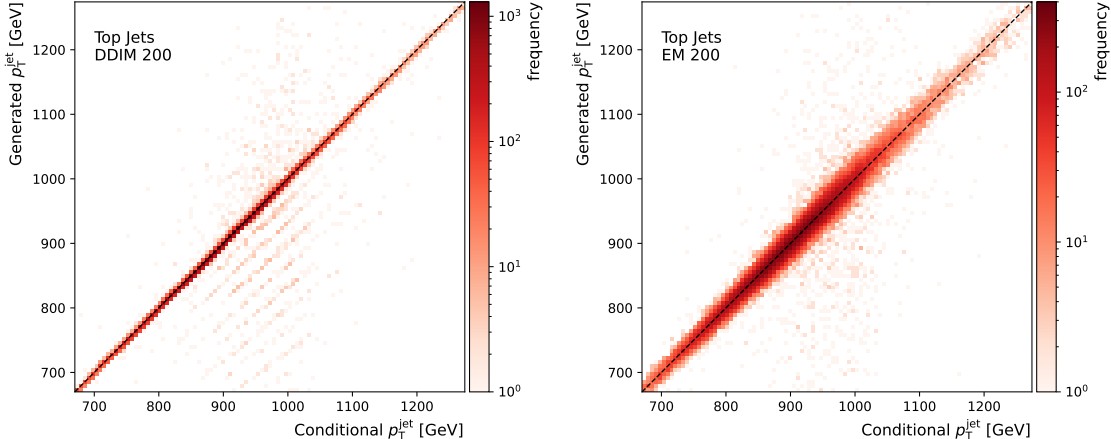

Figure 16: Two-dimensional histograms showing the correlation between the generated and conditional jet $p_\mathrm{T}$ for the top jets using the DDIM and EM solvers.

## 5.4 Timing comparison

To be used as fast surrogate models, the generation time required by a generative model is a crucial factor. MPGAN only requires one forward pass of the generator network, whereas PC-JeDi, as a diffusion model, requires several denoising steps. In Table 3 we compare the time required for one forward pass and the full generation for both models. Here, the choice of ODE/SDE solver in PC-JeDi plays a negligible role in generation time and therefore no distinction is made between them. We also look at the effect of batching the data and generating multiple events simultaneously using a GPU.

Table 3: Time required for generation for the MPGAN and PC-JeDi using either a CPU or a GPU. Generation times are calculated for a single forward pass through the network, as well as for the 200 integration steps required by PC-JeDi as a diffusion model. The times are also calculated for generation of a single jet, as well as batches of 10 and 1000 jets. The generation times are calculated using a single core of an AMD EPYC 7742 CPU and a single NVIDIA RTX 3080 GPU. The mean and standard deviations are calculated from 10 iterations. The required time for the jet simulation from the traditional MC simulation is taken from Ref. [33].

| Model | Hardware | Batch Size | # of forward calls | Time (ms) |
|---|---|---|---|---|
| MC Simulation | CPU | – | – | 46.2 |
| MPGAN | CPU | 1 | 1 | $4.83 \pm 0.04$ |
| | GPU | 1 | 1 | $3.52 \pm 0.22$ |
| | | 10 | | $4.88 \pm 0.08$ |
| | | 1000 | | $51.67 \pm 1.47$ |
| PC-JeDi | CPU | 1 | 1 | $5.98 \pm 0.18$ |
| | | | 200 | $1023.91 \pm 3.54$ |
| | GPU | 1 | 1 | $3.12 \pm 0.04$ |
| | | | 200 | $498.44 \pm 1.86$ |
| | | 10 | 1 | $3.30 \pm 0.12$ |
| | | | 200 | $515.76 \pm 1.05$ |
| | | 1000 | 1 | $24.48 \pm 0.30$ |
| | | | 200 | $4721.58 \pm 8.02$ |

Although the time required for a single pass through the network is similar between MPGAN and PC-JeDi for a single jet, the benefits of the transformer architecture become apparent as the number of jets in a batch increases. For a single jet, a single network pass takes approximately the same time, however for a batch size of 1000 we see that the transformer architecture requires half the time as the message passing graph layers in MPGAN. However, as diffusion models require multiple passes through the same network for generation, the time required to generate jets with PC-JeDi is $\mathcal{O}(100)$ greater than with MPGAN. With very large batch sizes, PC-JeDi averages around 4.72 ms per jet, which represents a speed up factor of $\mathcal{O}(10)$ compared to the time required for traditional MC generation.

## 6 Conclusion

In this work we present a novel conditional generative model for jets as particle clouds called PC-JeDi. The method follows a score-based formulation of diffusion processes and integration samplers which is highly customisable for future improvements. It is based on a permutation equivariant transformer architecture allowing it to naturally handle the point cloud structure of the data.

PC-JeDi is able to generate jet constituents with high fidelity, beating the state-of-the-art approach in several metrics. We also assessed additional substructure metrics not presented in the relevant literature so far, which we think are especially important for the downstream

physics applications. Furthermore, by studying two-dimensional marginals and uncontained top quark jets, we demonstrated that it is able to capture complex underlying correlations with conditional generation.

While generation quality is competitive with the current state-of-the-art method, the time required to generate samples with diffusion models is the main drawback of PC-JeDi. In addition, the ability for PC-JeDi to generate higher multiplicity particle clouds and the impact on generation speed and fidelity needs to be understood.

However, thanks to the flexibility of the method there are several avenues that can be considered to improve both the quality and the speed of the model. The conditional performance could be improved by the addition of auxiliary supervised regression loss terms during training, or by using more sophisticated guided diffusion techniques [87]. Furthermore, other network architectures could be explored, such as deep sets, which would reduce the time of a single pass through the network without a large trade-off in fidelity [36]. Moreover, one of the main areas of focus with diffusion models is the development of smarter, more efficient solvers. Within two years alone these models have gone from requiring $\mathcal{O}(1000)$ steps to generate high quality image data [44, 57], to $\mathcal{O}(100)$ [51], or even $\mathcal{O}(20)$ steps [39, 52]. Combining these two developments should improve the competitiveness of the generation speed of PC-JeDi and hopefully preserve the fidelity of generated jets.

# Acknowledgements

The authors would like to acknowledge funding through the SNSF Sinergia grant called "Robust Deep Density Models for High-Energy Particle Physics and Solar Flare Analysis (RODEM)" with funding number CRSII5_193716 and the SNSF project grant 200020_212127 called "At the two upgrade frontiers: machine learning and the ITk Pixel detector". They would also like to acknowledge the funding acquired through the Swiss Government Excellence Scholarships for Foreign Scholars and the Feodor Lynen Research Fellowship from the Alexander von Humboldt foundation.

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

## A  Training and generation algorithms

| **Algorithm 1** Training |
|---|
| **Require:** $\gamma(t), \sigma(t), L$ |
|   **while** not converged **do** |
|     $(\boldsymbol{x}, \boldsymbol{y}) \sim p_{\text{data}}$ |
|     $\boldsymbol{\epsilon} \sim \mathcal{N}(\boldsymbol{0}, \boldsymbol{I}), \, t \sim \mathcal{U}(0,1)$ |
|     $\boldsymbol{x} \leftarrow \text{Norm}(\boldsymbol{x})$ |
|     $\boldsymbol{y} \leftarrow \text{Norm}(\boldsymbol{y})$ |
|     $\boldsymbol{v}_t \leftarrow \text{CosEnc}(t)$ |
|     $\boldsymbol{x}_t \leftarrow \gamma(t)\boldsymbol{x} + \sigma(t)\boldsymbol{\epsilon}$ |
|     Optimise: $L(\boldsymbol{\epsilon}, \hat{\boldsymbol{\epsilon}}_\theta(\boldsymbol{x}_t, \boldsymbol{v}_t, \boldsymbol{y}))$ |
|   **end while** |

| **Algorithm 2** Generation |
|---|
| **Require:** $\gamma(t), \sigma(t), \boldsymbol{y}, \phi, \Delta t$ |
|   $\boldsymbol{x}_t \sim \mathcal{N}(\boldsymbol{0}, \boldsymbol{I}), \, t \leftarrow 1$ |
|   $\boldsymbol{y} \leftarrow \text{Norm}(\boldsymbol{y})$ |
|   **while** $t > 0$ **do** |
|     $\boldsymbol{v}_t \leftarrow \text{CosEnc}(t)$ |
|     $\boldsymbol{x}_t \leftarrow \phi(\boldsymbol{x}_t, \hat{\boldsymbol{\epsilon}}_\theta(\boldsymbol{x}_t, \boldsymbol{v}_t, \boldsymbol{y}), t)$ |
|     $t \leftarrow t - \Delta t$ |
|   **end while** |
|   $\boldsymbol{x}_0 \leftarrow \text{UnNorm}(\boldsymbol{x}_t)$ |
|   **return** $\boldsymbol{x}_0$ |

## B  Network setup and hyperparameters

The TE block used in PC-JeDi is based on the *Normformer* [88] encoder block. It is depicted in Fig. 17. The block is composed of a residual attention network followed by a residual dense network. The attention network takes the point cloud as input tokens and performs a multi-headed self-attention pass surrounded by layer normalisations. The intermediate tokens are then added to the input tokens via a residual connection. The context properties $\boldsymbol{c}$, which in our case are the encoded time vector, jet mass, and jet $p_{\text{T}}$, are concatenated to the features of each individual token before being processed by the dense network. The dense network comprises two fully connected linear layers. A sigmoid-linear-unit (SiLU) activation is applied to the output of the hidden layer, layer normalisation is used to keep the gradients stable, and dropout, as this is a supervised setting, is used for regularization. The output tokens are then added to the intermediate tokens via another residual connection. The input and output dimensions of the token features are the same, so several entire TE-Blocks can be chained together.

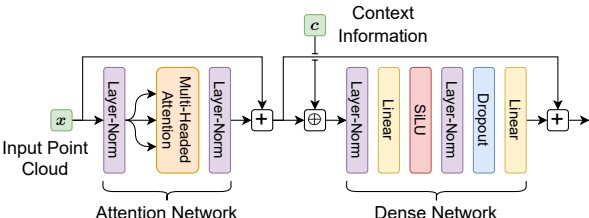

Figure 17: Our Transformer-Encoder block is made of a residual self-attention network followed by a residual dense network. Context information is concatenated to the intermediate tokens before they are passed to the dense network.

After running a hyperparameter grid search separately for top jets and gluon jets, the model architecture which minimised the validation loss in each case comprises four transformer encoder blocks each with a base dimension size of 128. Each dense network has a single hidden layer of size 256 and the dropout rate is set to 0.1. For the time embedding layer we use cosine encoding with an output size of $M = 16$.

We use the Adam optimizer [89] with default settings, a batch size of 256, and set the learning rate to ramp up linearly from 0 to 0.0005 over the first 10000 training iterations. The best network is selected based on the minimum value of the loss on the validation dataset. FPND and $W_1^{\text{M}}$ values for jets generated using the EM sampler with 100 steps were tracked

alongside the validation loss. They were found to follow the same trend as the validation loss, and so only the latter was used for optimisation. For the plots and figures shown in Section 5, we generate 50000 jets using conditional information sampled from the test set.

The input representations were also studied in a grid search, both for the constituent four momenta and the jet conditional variables. PC-JeDi was trained both with and without conditional information. Using the invariant mass and transverse momentum of the jet calculated using all constituents and not just the leading 30 constituents was also studied. A negligible impact on performance was observed when comparing these options. The values calculated using only the leading 30 constituents were chosen in order to test for the diagonality of the generated jet momenta and invariant masses. Whether to use the transverse momentum or relative transverse momentum of the constituents was also studied in conjunction with $\log(p_T)$ and $\log(p_T + 1)$ transformations. Using $\log(p_T + 1)$ for the constituent momenta was found to perform best in most metrics, and resulted in better agreement for low $p_T$ constituents.

## C   Integration Samplers

### C.1   Detailed integration sampler algorithms

We detail here the four integration sampler algorithms used in our experiments for solving the differential equations. All of them use a parametrised noise estimator $\hat{\epsilon}_\theta$ which is conditioned on the perturbed input $x_t$, the diffusion time $t$ and any other conditional variables $y$. However, in our study this noise estimator is a neural network which takes a cosine encoded time $v_t$ instead of $t$. The reader will make the correspondence accordingly when comparing to the generic integration step $\phi$ used in the generation procedure shown in Fig. 2. The variance preserving SDE expressed in Eq. (3) is integrated using the Euler-Maruyama solver shown in Algorithm 3. For this procedure, the integration step $x_t \leftarrow \phi(x_t, \hat{\epsilon}_\theta, t)$ corresponds to Line 3 to Line 5. On the other hand, for the DDIM reverse diffusion solver shown in Algorithm 4 the integration step corresponds to Line 3 to Line 6. Note that the time update $t \leftarrow t - \Delta t$ for DDIM uniquely takes place before the $x_t$ update.

---

**Algorithm 3**
Euler-Maruyama solver for VP SDE

---

**Require:** $N$, $\beta(t)$, $\sigma(t)$, $y$
1: $\Delta t \leftarrow \frac{1}{N}$, $t \leftarrow 1$, $x_t \sim \mathcal{N}(0, I)$
2: **while** $t > 0$ **do**
3: $\quad x_t \leftarrow x_t + \frac{1}{2}\beta(t)\left[x_t - 2\frac{\hat{\epsilon}_\theta(x_t, t, y)}{\sigma(t)}\right]\Delta t$
4: $\quad z \sim \mathcal{N}(0, I)$
5: $\quad x_t \leftarrow x_t + \sqrt{\beta(t)\Delta t}\, z$
6: $\quad t \leftarrow t - \Delta t$
7: **end while**
8: **return** $x_t$

---

**Algorithm 4**
DDIM solver for reverse diffusion

---

**Require:** $N$, $\beta(t)$, $\sigma(t)$, $\gamma(t)$, $y$
1: $\Delta t \leftarrow \frac{1}{N}$, $t \leftarrow 1$, $x_t \sim \mathcal{N}(0, I)$
2: **while** $t > 0$ **do**
3: $\quad \hat{\epsilon} \leftarrow \hat{\epsilon}_\theta(x_t, t, y)$
4: $\quad \hat{x}_0 \leftarrow \frac{x_t - \sigma(t)\hat{\epsilon}}{\gamma(t)}$
5: $\quad t \leftarrow t - \Delta t$
6: $\quad x_t \leftarrow \gamma(t)\hat{x}_0 + \sigma(t)\hat{\epsilon}$
7: **end while**
8: **return** $\hat{x}_0$

---

The variance preserving ODE expressed in Eq. (4) is integrated using two different solvers. For the Euler solver shown in Algorithm 5 the integration step $x_t = \phi(x_t, \hat{\epsilon}_\theta, t)$ corresponds to Line 3. The Runge-Kutta fourth order solver shown in Algorithm 6 is slightly more involved since it requires four network evaluations. Therefore, the integration step must be understood as the whole block of Line 3 to Line 7, the four network evaluations being done with the properly shifted $x_t$ and $t$. Notice that ignoring $k_2$, $k_3$ and $k_4$ would lead to the Euler solver.

**Algorithm 5**
Euler solver for VP ODE

**Require:** $N, \beta(t), \sigma(t), \mathbf{y}$
1: $\Delta t \leftarrow \frac{1}{N}, t \leftarrow 1, \mathbf{x}_t \sim \mathcal{N}(\mathbf{0}, \mathbf{I})$
2: **while** $t > 0$ **do**
3: $\quad \mathbf{x}_t \leftarrow \mathbf{x}_t + \frac{1}{2}\beta(t)\left[\mathbf{x}_t - \frac{\hat{\boldsymbol{\epsilon}}_\theta(\mathbf{x}_t, t, \mathbf{y})}{\sigma(t)}\right]\Delta t$
4: $\quad t \leftarrow t - \Delta t$
5: **end while**
6: **return** $\mathbf{x}_t$

**Algorithm 6**
Runge-Kutta $4^{\text{th}}$ order solver for VP ODE

**Require:** $N, \beta(t), \sigma(t), \mathbf{y}$
1: $t \leftarrow 1, \Delta t \leftarrow \frac{1}{N}, \mathbf{x}_t \sim \mathcal{N}(\mathbf{0}, \mathbf{I})$
2: **while** $t > 0$ **do**
3: $\quad \mathbf{k}_1 \leftarrow \frac{1}{2}\beta(t)\left[\mathbf{x}_t - \frac{\hat{\boldsymbol{\epsilon}}_\theta(\mathbf{x}_t, t, \mathbf{y})}{\sigma(t)}\right]\Delta t$
4: $\quad \mathbf{k}_2 \leftarrow \frac{1}{2}\beta(t - \frac{\Delta t}{2})\left[\mathbf{x}_t - \frac{\hat{\boldsymbol{\epsilon}}_\theta(\mathbf{x}_t + \frac{\mathbf{k}_1}{2}, t - \frac{\Delta t}{2}, \mathbf{y})}{\sigma(t - \frac{\Delta t}{2})}\right]\Delta t$
5: $\quad \mathbf{k}_3 \leftarrow \frac{1}{2}\beta(t - \frac{\Delta t}{2})\left[\mathbf{x}_t - \frac{\hat{\boldsymbol{\epsilon}}_\theta(\mathbf{x}_t + \frac{\mathbf{k}_2}{2}, t - \frac{\Delta t}{2}, \mathbf{y})}{\sigma(t - \frac{\Delta t}{2})}\right]\Delta t$
6: $\quad \mathbf{k}_4 \leftarrow \frac{1}{2}\beta(t - \Delta t)\left[\mathbf{x}_t - \frac{\hat{\boldsymbol{\epsilon}}_\theta(\mathbf{x}_t + \mathbf{k}_3, t - \Delta t, \mathbf{y})}{\sigma(t - \Delta t)}\right]\Delta t$
7: $\quad \mathbf{x}_t \leftarrow \mathbf{x}_t + \frac{\mathbf{k}_1 + 2\mathbf{k}_2 + 2\mathbf{k}_3 + \mathbf{k}_4}{6}$
8: $\quad t \leftarrow t - \Delta t$
9: **end while**
10: **return** $\mathbf{x}_t$

## C.2 Choice of samplers

To understand the trade-off between the quality of the generated samples and the generation speed, we test four different methods for sample generation using the same trained model. These different methods, or solvers, are labelled: DDIM, Euler-Maruyama (EM), Euler and Runge-Kutta (RK). We study the effect of the number of integration steps, or network passes, on the samples quality using the metrics introduced in Section 4.4. We focus on two metrics for brevity: Coverage (Cov), which is indicative of the diversity of the generated jets compared to MC, and the Wasserstein-1 distance between the generated and real jet-mass distributions $\left(W_1^M\right)$.

Figure 18 shows that the generation quality increases with a larger number of iteration steps irrespective of the choice of ODE/SDE solver. However, the difference between the solvers becomes negligible for these metrics beyond around 100 network passes compared to run variations. We also notice little improvement in the quality of the generated jets beyond 200 network passes for all solvers.

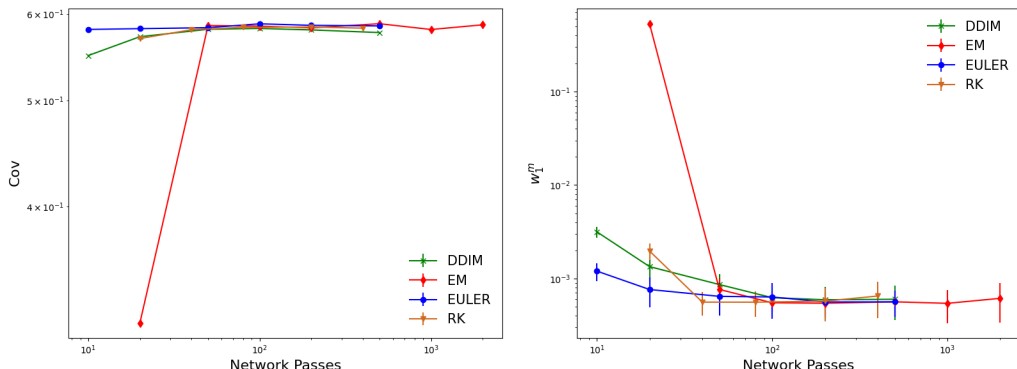

Figure 18: Cov (higher is better) and $W_1^M$ (lower is better) metrics verses the number of network passes used in the full generation for four different solvers on the top jet dataset. DDIM (green), EM (red), Euler (blue), RK (orange) all saturate near $\mathcal{O}(100)$ network passes.

# D  Additional figures and tables

## D.1  Relative constituent coordinates

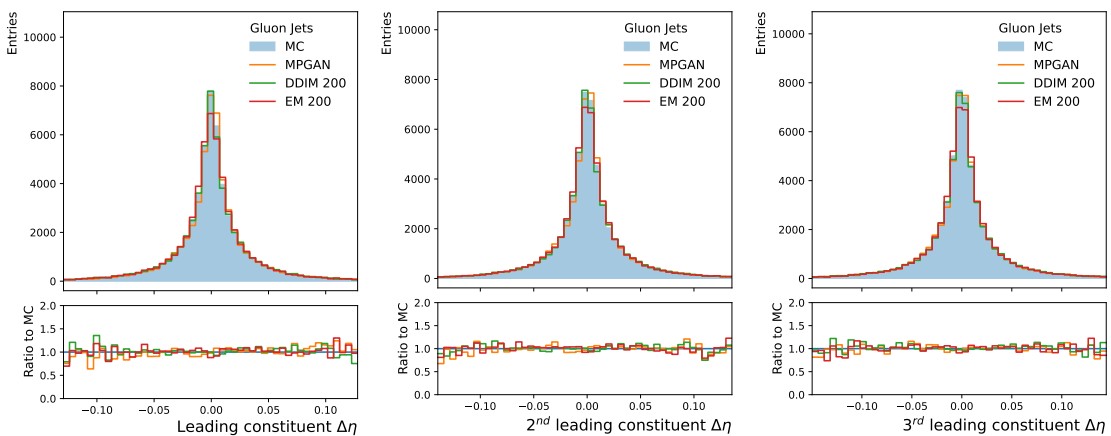

Figure 19: Distributions of constituent $\Delta\eta$ for the gluon jets.

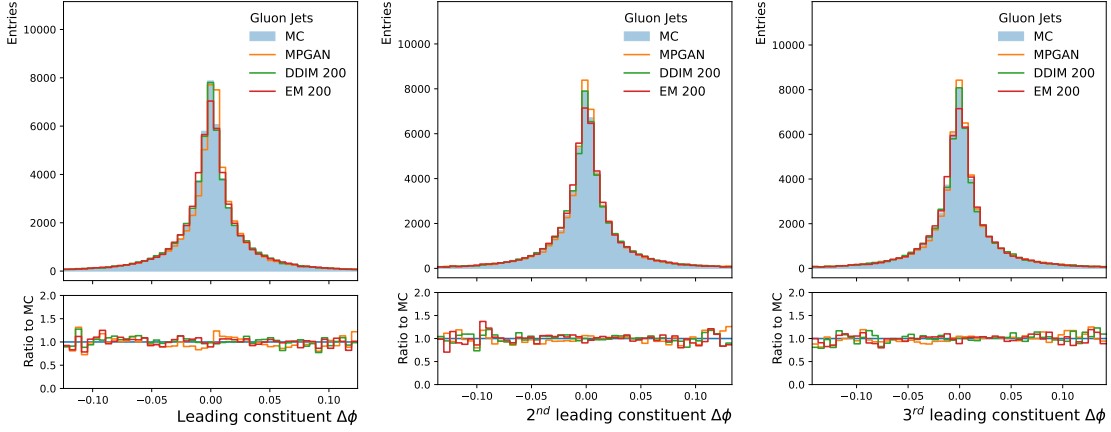

Figure 20: Distributions of constituent $\Delta\phi$ for the gluon jets.

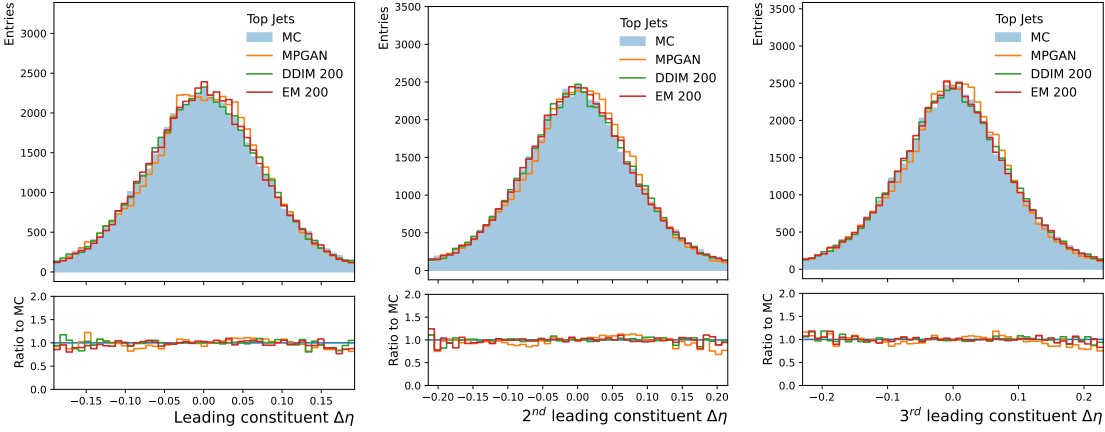

Figure 21: Distributions of constituent $\Delta\eta$ for the top jets.

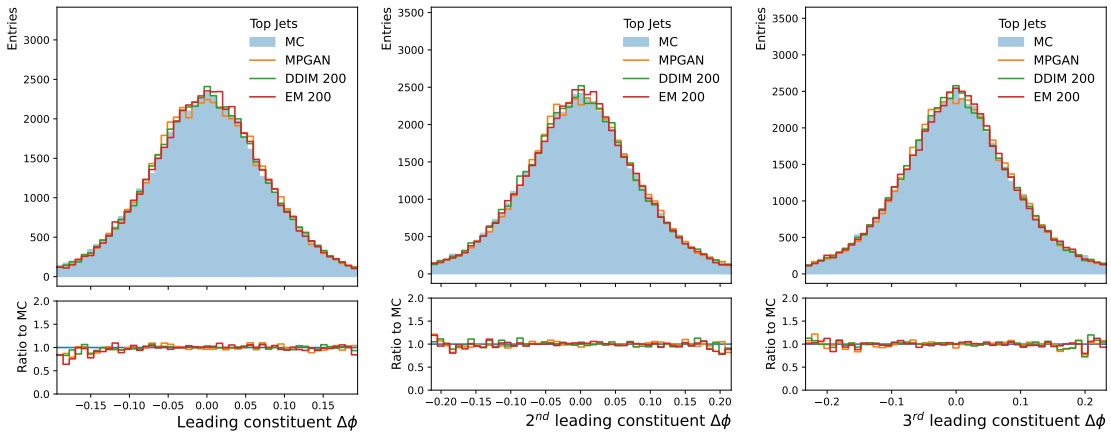

Figure 22: Distributions of constituent $\Delta\phi$ for the top jets.

## D.2 Comparison of samplers

Table 4: Comparison of diffusion sampling methods for the metrics introduced in Ref. [33]. Lower is better for all metrics except Cov. Note that these metrics were taken using different sample sizes and thus are not directly comparable with the values from in Table 1.

| Jet Class | Sampler (steps) | $W_1^M(\times10^{-3})$ | $W_1^P(\times10^{-3})$ | $W_1^{EFP}(\times10^{-5})$ | FPND | Cov ↑ | MMD $(\times10^{-2})$ |
|---|---|---|---|---|---|---|---|
| Gluon | DDIM (200) | 0.78 ± 0.34 | 0.98 ± 0.53 | 0.83 ± 0.71 | 1.55 | **0.55** | **3.37** |
| | EM (200) | **0.53 ± 0.19** | **0.61 ± 0.27** | **0.61 ± 0.53** | 1.45 | 0.55 | 3.38 |
| | Euler (200) | 0.61 ± 0.22 | 0.81 ± 0.38 | 0.63 ± 0.48 | 1.47 | 0.55 | 3.41 |
| | RK (50) | 0.56 ± 0.22 | 0.77 ± 0.35 | 0.71 ± 0.52 | 1.49 | **0.55** | 3.38 |
| Top | DDIM (200) | 0.59 ± 0.22 | **0.63 ± 0.35** | 2.57 ± 1.72 | 0.68 | **0.58** | 6.50 |
| | EM (200) | **0.54 ± 0.15** | 0.99 ± 0.44 | 1.51 ± 1.28 | 0.38 | 0.58 | 6.48 |
| | Euler (200) | 0.56 ± 0.17 | 0.74 ± 0.43 | **1.33 ± 1.06** | 0.49 | **0.59** | 6.47 |
| | RK (50) | 0.57 ± 0.22 | 0.80 ± 0.37 | 1.35 ± 1.04 | 0.51 | 0.58 | 6.47 |

Table 5: Comparison of diffusion sampling methods for the substructure derived metrics.

| Jet Class | Sampler (steps) | $W_1^{\tau_1}(\times10^{-3})$ | $W_1^{\tau_2}(\times10^{-3})$ | $W_1^{\tau_3}(\times10^{-3})$ | $MAE^M(\times10^{-2})$ | $MAE^{P_T}(\times10^{-2})$ |
|---|---|---|---|---|---|---|
| Gluon | DDIM (200) | **0.79 ± 0.30** | 2.38 ± 0.18 | 1.82 ± 0.11 | **0.05** | **0.44** |
| | EM (200) | 0.87 ± 0.27 | 0.92 ± 0.15 | **0.54 ± 0.07** | 0.10 | 1.29 |
| | Euler (200) | 1.09 ± 0.29 | 1.20 ± 0.16 | 0.70 ± 0.07 | 0.10 | 1.23 |
| | RK (50) | 1.01 ± 0.24 | 1.30 ± 0.17 | 0.75 ± 0.08 | 0.10 | 1.23 |
| DDIM (200) | | **0.66 ± 0.19** | 3.37 ± 0.34 | 3.96 ± 0.13 | **0.06** | **0.44** |
| | EM (200) | 0.72 ± 0.22 | **0.93 ± 0.30** | 1.07 ± 0.08 | 0.19 | 1.24 |
| | Euler (200) | 0.78 ± 0.27 | 1.74 ± 0.38 | 1.48 ± 0.12 | 0.18 | 1.18 |
| | RK (50) | 0.70 ± 0.19 | 2.04 ± 0.36 | 1.67 ± 0.11 | 0.18 | 1.18 |

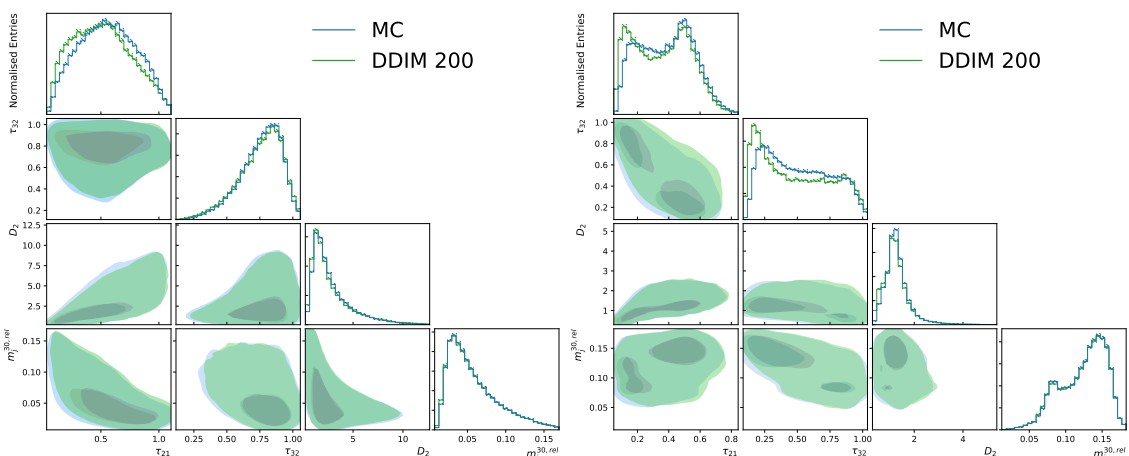

Figure 23: Mass and relative substructure distributions of the generated gluon (left) and top (right) jets using the DDIM solver. The diagonal consists of the marginals of the distributions. The off-diagonal elements are the joint distributions of the variables.

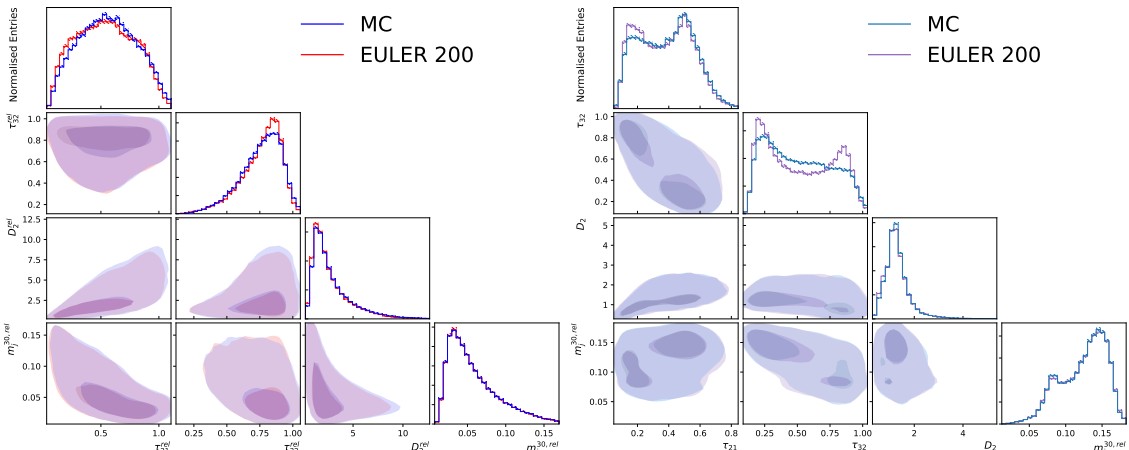

Figure 24: Mass and relative substructure distributions of the generated gluon (left) and top (right) jets using the Euler solver. The diagonal consists of the marginals of the distributions. The off-diagonal elements are the joint distributions of the variables.

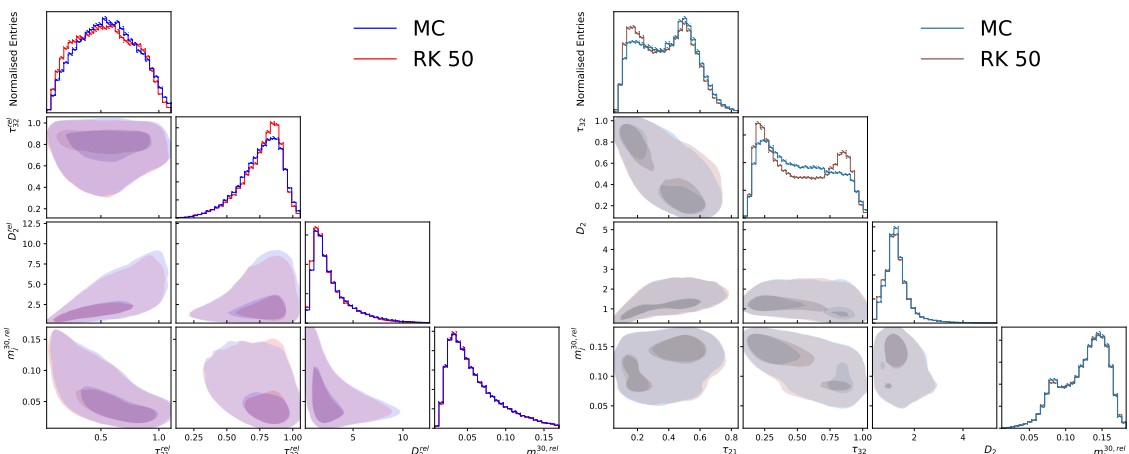

Figure 25: Mass and relative substructure distributions of the generated gluon (left) and top (right) jets using the Runge-Kutta solver. The diagonal consists of the marginals of the distributions. The off-diagonal elements are the joint distributions of the variables.