# Peer review of "PC-JeDi: Diffusion for Particle Cloud Generation in High Energy Physics"

_SciPost Physics_

## Round 1 · Referee Report · Anonymous (Referee 1) · 2023-8-10

Strengths

1- Innovative method for jet generation. 2- Strong performance compared to standard baselines. 3- Authors provide additional jet substructure observables to highlight the capability of the model to learn complex correlations between particles.

Weaknesses

1 - Lack of details about the individual choice of parameters in the main text (beta, alpha) 2- Unclear motivation for some design choices (modified loss function with additional alpha parameter)

Report

The authors introduce PC-JeDi, a diffusion-based model for jet generation. The proposed model shows remarkable performance at generating jets from quarks and gluons, often achieving state-of-the-art performance across different metrics. The work is novel and I recommend for publication in SciPost Physics.

The authors should make the choice of parameters explicit in the main body of the text instead of describing the details in an appendix. More notably, Eq.7 that describes the main loss function, is misleading assuming that later the authors choose a different norm (Huber loss) for the training. On the same note, the choice of mixing MLE and standard diffusion training in the loss function does not seem to be well motivated. The authors point out the instability with \sigma(t) when t→0, which is still used in the loss function. How does the parameter \alpha gets to be chosen? How much the results change if \alpha=0?
The results are presented clearly with good number of evaluation metrics to test the performance of the generative model. The authors also point out that the truncation of the number of particles leads to a mismatch between generated jet kinematics and true jet kinematics (pT and mass). In the conditional generation however, the authors show an almost diagonal match between reconstructed and conditional inputs. Are the conditional quantities used during training derived from the jets after truncation or do they represent the original jet kinematics?

Requested changes

1- Fix the choice of MLE weighting \lambda(t) = \beta(t) and not \beta(t)^2 in the text and Eq.7 2- Explicitly mention the choice of parameters used as they are introduced in the text: beta parameterization and upper boundary used in Eq. 2, choice of alpha in Eq.7 , resulting choice of \sigma(t) and \gamma(t). 3- Directly after Eq.7 to mention the use of Huber loss instead of MSE or add that directly to the loss function. 4- Add an explanation on how \alpha is chosen and the impact of this choice on the results presented. Good but not necessary: comparison of the results when \alpha=0 or at least a comment on the impact of this choice in the results presented.

---

## Round 1 · Referee Report · Anonymous (Referee 2) · 2023-9-8

Strengths

  1. The manuscript describes the application of a novel generative model to a well-motivated physics problem. It is well structured and shows a very promising performance of the proposed model.

  2. The authors evaluate their model using a variety of metrics. They also look at substructure observables and correlations.

Weaknesses

  1. There are a few places in which the explanations could be improved, or more details could be given.

Report

My apologies for the delay in providing this report. This manuscript discusses the application of a diffusion-based generative model to the simulation of jets at the LHC, using a point cloud representation. The problem is well motivated and described. The proposed solution shows very promising results, also when compared to the an alternative generative model (the MPGAN).

I would definitively recommend this manuscript for publication in SciPost Physics, but before I would like to see a few minor improvements in discussions and presentation, see my list below.

  • There are a few more references that used point clouds for jet physics, for example 2102.05073. Maybe the authors could check if they missed more related literature and expand section 2. The HEP-ML living review could be helpful for this.
  • For the comparison to the MPGAN, did you train your own MPGAN? Was the performance of that as in the respective paper?
  • How does equation 5 link to the selection made on page 25? Inserting the definition of $\beta$, I cannot reproduce $\sigma$ and $\gamma$.
  • In several places (for example in the timing section), the authors discuss the 'speed of inference'. I think 'speed of generation' would be a better description for this. (In other setups, like normalizing flows, 'inference' and 'generation' refer to different things with different timings and that might confuse the reader here, since it is really about the time needed for generation.)
  • The architecture is said to be permutation equivariant in section 4.3. However, figs. 1 and 2 as well as the text on p. 8 say that the point cloud is passed through a dense layer before entering the TE-blocks. Is that a small dense layer for all jet constituents (like in a deep set)? Or is that a large dense layer taking in all constituents, therefore breaking the permutation equivariance?
  • What is $e$ in the $M$-dimensional time-encoding vector $\nu_t$?
  • How Gaussian is the distribution in the penultimate layer of ParticleNet? Are the assumptions of FPND fulfilled?
  • How stable are the results in tables 1 and 2? How much do the values change when a new sample is generated from the same generator? How much do they change when the generator is re-trained? Where do the error-bars for $W_1$ come from? Please estimate the errors for the other metrics.
  • I don't understand the sentence "Although the time required for a single pass through the network is similar between MPGAN and PC-JeDi for a single jet, the benefits of the transformer architecture become apparent as the number of jets in a batch increases." Isn't the MPGAN as a GAN also able to generate batches?
  • What is the timing for the standard MC generation for the jets (as a reference for table 3).
  • For the RK solver in Appendix C2, are the number of NN passes and the number of integration steps the same, or do they differ by a factor of 4 because lines 3-6 of algorithm 6 each call the NN once? Does that mean that the number of integration steps in fig. 18 do not match between methods?
  • How many runs are the error bars of fig. 18 based on?
  • As an optional suggestion, since it came out after this manuscript: The authors mention it could be worth looking at a metric that is more sensitive to tails on p. 13, would a classifier-based weight study as suggested in 2305.16774 be an option?

Requested changes

Please address the questions in my report.

---

## Editorial Decision

resubmitted